# Altered Cytokine Response of Human Brain Endothelial Cells after Stimulation with Malaria Patient Plasma

**DOI:** 10.3390/cells10071656

**Published:** 2021-07-01

**Authors:** Michaela Raacke, Amy Kerr, Michael Dörpinghaus, Jana Brehmer, Yifan Wu, Stephan Lorenzen, Christine Fink, Thomas Jacobs, Thomas Roeder, Julie Sellau, Anna Bachmann, Nahla Galal Metwally, Iris Bruchhaus

**Affiliations:** 1Bernhard Nocht Institute for Tropical Medicine, 20359 Hamburg, Germany; michaela.raacke@web.de (M.R.); a.kerr@freenet.de (A.K.); michael.doerpinghaus@web.de (M.D.); jana.brehmer@bnitm.de (J.B.); wu.yifan@bnitm.de (Y.W.); lorenzen@bnitm.de (S.L.); tjacobs@bnitm.de (T.J.); sellau@bnitm.de (J.S.); bachmann@bnitm.de (A.B.); metwally@bnitm.de (N.G.M.); 2Department of Molecular Physiology, Kiel University, 24118 Kiel, Germany; cfink@zoologie.uni-kiel.de (C.F.); troeder@zoologie.uni-kiel.de (T.R.); 3Airway Research Center North (ARCN), Member of the German Center for Lung Research (DZL), 24118 Kiel, Germany; 4Department of Biology, University of Hamburg, 20148 Hamburg, Germany

**Keywords:** *Plasmodium falciparum*, malaria, endothelial cells, cytokines

## Abstract

Infections with the deadliest malaria parasite, *Plasmodium falciparum*, are accompanied by a strong immunological response of the human host. To date, more than 30 cytokines have been detected in elevated levels in plasma of malaria patients compared to healthy controls. Endothelial cells (ECs) are a potential source of these cytokines, but so far it is not known if their cytokine secretion depends on the direct contact of the *P. falciparum*-infected erythrocytes (IEs) with ECs in terms of cytoadhesion. Culturing ECs with plasma from malaria patients (27 returning travellers) resulted in significantly increased secretion of IL-11, CXCL5, CXCL8, CXCL10, vascular endothelial growth factor (VEGF) and angiopoietin-like protein 4 (ANGPTL4) if compared to matching controls (22 healthy individuals). The accompanying transcriptome study of the ECs identified 43 genes that were significantly increased in expression (≥1.7 fold) after co-incubation with malaria patient plasma, including *cxcl5* and *angptl4*. Further bioinformatic analyses revealed that biological processes such as cell migration, cell proliferation and tube development were particularly affected in these ECs. It can thus be postulated that not only the cytoadhesion of IEs, but also molecules in the plasma of malaria patients exerts an influence on ECs, and that not only the immunological response but also other processes, such as angiogenesis, are altered.

## 1. Introduction

Despite the advances in malaria control programs, malaria remains one of the most detrimental infectious diseases worldwide. In 2019, about 229 million cases of malaria were recorded, including 409,000 cases of deaths [1]. Within the five species that cause malaria in humans, *Plasmodium falciparum* is the most clinically relevant one and responsible for most deaths. The complications caused by malaria infection are multifactorial; both the parasite and the host contribute. A central part of the pathogenesis is the cytoadhesion of *P. falciparum*-infected erythrocytes (IEs) within the vascular bed of vitally important organs, such as the brain, heart, lung, stomach, skin and kidney [2,3,4]. Besides blockage of capillaries due to the cytoadhesion of the IEs, increased inflammatory cytokine production, endothelial dysfunction and increased vascular permeability also occur in the affected tissue [5,6,7]. As a result of the immune response induced by parasite growth and cytoadhesion to the endothelium, patients develop fever, headaches, muscle aches and rigors [8,9,10,11,12]. According to the age and immune status of the patient, severe lethal complications, such as cerebral malaria (CM), lung injury, renal failure, acidosis and severe anaemia, may develop [11,12,13]. Not only the acute complications affect the patients, but also one third of survivors (African children and adult travellers) were found to suffer from long-term health problems, such as cognitive and neurological impairments [14,15,16].

Several in vitro studies have shown that interaction of the IEs with endothelial cells (ECs) increase the expression of various genes encoding proinflammatory cytokines, such as IL-6, IL-1 and tumor necrosis factor-alpha (TNF-α), and chemokines, such as CXCL8, CCL2, CCL20, CXCL1, CXCL2 and CCL5, which are important for the recruitment of leukocytes to the endothelium during inflammation [17,18,19]. 

Diverse plasmodial antigens and released components, as well as endogenous metabolites associated with danger signals, have repeatedly been shown to stimulate the immune system. Glycosylphosphadidylinositol anchors of *P. falciparum* proteins have been the first ones described [20], leading to production of the proinflammatory cytokines in macrophages [21]. It was also shown that in monocytes fed with hemozoin, expression of genes encoding for cytokines and chemokines was increased [22]. In macrophages, incubation with hemozoin resulted in an increase in various chemokine transcripts, including CCL3, CCL4, CXCL1 and CCL2 [23], but phagocytosis of hemozoin leads to impairment of macrophage function [24]. Thus, *P. falciparum* malaria is accompanied by a strong immunological reaction of the host. This considerable increase in systemic and local inflammation contributes significantly to the pathogenesis of malaria. The immune response triggered by *P. falciparum* is a very complex event. To date, well over 30 cytokines have been described, which can be detected in serum or plasma of malaria patients in larger quantities compared to healthy controls. These include CCL2, CCL4, CXCL4, CXCL8, CXCL10, CCL3, IL-1β, IL-6, CXCL2, TNF-α, interferon-gamma (INF-γ), IL-1α, IL-12, IL-17A, IL-15, IL-10, IL1-RA, CCL20, vascular endothelial growth factor (VEGF), IL-13, IL-31, IL-33, CXCL9, IL-9, CCL28 and granulocyte-colony stimulating factor (G-CSF) [5,25,26,27,28,29,30,31,32,33,34,35,36,37,38]. Significantly increased amounts of CCL2, CCL4, CXCL4, CXCL8, CXCL10, IL-1RA, IL-6, TNF-α and G-CSF were detected in patients with CM [27,29,39,40] and the amount of some cytokines (CCL2, CCL4, CXCL4, CXCL8, CXCL10 and CCL3) directly correlates with the severity of a malaria infection [25,26,29,30,36].

Plasma levels of 29 biomarkers, including various chemokines and cytokines, were investigated in patients with CM and non-CM. However, significantly increased levels in patients with CM compared to non-CM were only found for IL-6, CXCL8 and IL-1RA [26]. A positive correlation with parasitaemia was described for CCL20 and CXCL9 [36].

To date, there are no studies comparing the levels of different cytokines in the plasma of *P. falciparum*-infected returning travellers and, in parallel, elucidating the influence of these plasmas on the stimulation of ECs and the associated secretion of these factors into the cell culture supernatant as well as their influence on EC gene expression. 

Our studies showed that co-incubation of brain ECs with plasma from malaria patients resulted in significantly increased secretion of IL-11, CXCL5, CXCL10, VEGF and angiopoietin-like protein 4 (ANGPTL4). A comparative transcriptome analysis revealed that, in addition to an inflammatory response, biological processes such as cell migration, cell proliferation and tube development, as well as the KEGG ‘IL-17 signalling pathway’, were particularly affected. 

## 2. Materials and Methods

### 2.1. Blood Plasma of Malaria Patients and Healthy Control Individuals 

The study was performed on 27 EDTA-plasma samples from patients diagnosed with *P. falciparum* malaria, with parasitaemia between <1% and 11%. All patients were adult tropical returnees and were treated as in- or outpatients in Hamburg, Germany. Patients were either seen in the outpatient clinic of the University Medical Center Hamburg-Eppendorf (UKE) at the Bernhard Nocht Institute for Tropical Medicine, treated as inpatients at the UKE, or at the Bundeswehrkrankenhaus Hamburg. As controls, 22 plasma samples from healthy individuals were used. The study was approved by the relevant ethics committee (Ethical Review Board of the Medical Association of Hamburg, reference numbers PV3828 and PV4539) (Appendix A).

### 2.2. HBEC-5i Brain Endothelial Cell Line 

This project was carried out using human brain endothelial cells HBEC-5i, derived from the cerebral cortex and immortalized with the SV40 large T antigen (American Type Culture Collection (ATCC), Manassas, VA, USA; no. CRL-3245). HBEC-5i cells were seeded in 0.1% gelatin-coated T25 culture flasks. For normal cell culture, DMEM/F-12 complete growth medium (Gibco, Thermo Fisher Scientific, Bremen, Germany) containing 40 µg/mL endothelial cell growth supplement (ECGS; Merck Millipore, Darmstadt, Germany), 10% heat-inactivated foetal calf serum (Capricorn Scientific, Ebsdorfergrund, Germany) and 9 µg/mL gentamycin (Sigma–Aldrich Merck, Darmstadt, Germany) was used. The endothelial cells (ECs) were cultivated at 37 °C and 5% CO_2_ atmosphere and split every 2–4 days when a confluence of 70–90% is reached. 

### 2.3. Stimulation Assay of ECs with Plasma of Malaria Patients and Healthy Control Individuals

The 96-well plates were coated with 50 μL of 0.1% gelatin (Sigma–Aldrich Merck, Darmstadt, Germany) in Dulbecco’s Phosphate-Buffered Saline (DPBS; PAN, Biotech, Germany) per well and incubated at 37 °C for 30 min. After incubation, the gelatin was aspirated and 50 μL DMEM/F-12 medium was placed in each well and incubated at 37 °C for 15 min to adjust the pH value. After removal of the DMEM/F-12 medium, 1 × 10^4^ ECs in 200 µL DMEM/F-12 medium were added to each well. The cells were cultivated for two days with a medium change after the first day.

For the stimulation assay, the cells were washed twice with 100 µL/well DMEM/F-12 medium each before addition of the human plasma. In total, 80 µL of a plasma mixture consisting of 58 µL DMEM/F-12/gentamycin medium, 2 µL heparin (10,000 units/mL; Braun, Melsungen, Germany) and 20 µL human plasma were added per well. Each plasma sample was analysed in quadruple. The 96-well plate was then incubated for 6 h at 37 °C (5% CO_2_). After completion of the 6 h incubation, the supernatant was removed, and the wells were washed 4 times with DMEM/F-12/gentamycin medium. Then 100 µL of DMEM/F-12 complete growth medium was added and the cells were incubated for another 42 h before the cell culture supernatant was removed; after a total amount of 48 h, the four replicates were pooled, centrifuged and the supernatant immediately frozen at −80 °C.

For the transcriptome analyses, the ECs were incubated in T25 cell culture flasks (monolayer 70–90%) containing 4.5 mL DMEM/F-12/gentamycin medium, 50 µL heparin (10,000 units/mL; Braun, Melsungen, Germany) and 500 µL plasma of malaria patients and healthy controls, respectively (plasma concentration 10%), for 7 h. Afterwards, the cells were washed and lysed with 200 µL Trizol (Invitrogen, Thermo Fisher Scientific, Bremen, Germany) and stored at −80 °C until the RNA was isolated.

### 2.4. LEGENDplex^TM^ Assay 

The LEGENDplex Kits used were multiplex bead-based assay panels manufactured by BioLegend, Inc. (San Diego, CA, USA). The two bead panels that were chosen for measurement of cytokine concentration in every sample included the pro-inflammatory cytokines IL-1α and IL-1β, the pro- and anti-inflammatory cytokines IL-6, IL-7, IL-12 and IFN-β, the anti-inflammatory cytokines IL-1RA, IL-10 and IL-11, and the proinflammatory chemokines CCL3, CCL20, CXCL1, CXCL5, CXCL8, CXCL10 and VEGF. The bead-assays were performed following instructions provided by the manufacturer in duplicates. After completion of the reaction, the samples were transferred to FACS tubes to be read on a flow cytometer (BD Accuri^®^ C6 Flow Cytometer, Thermo Fisher Scientific, Bremen, Germany). 

The concentration of a particular analyte was determined by the provided LEGENDplexTM Software v8 based on a known standard curve. Values with evident methodical errors were excluded. After calculating the mean of the two replicated values for each analyte, statistical analyses were performed using GraphPad Prism (version 9.02 (134) GraphPad Software Inc, San Diego, CA, USA). A Mann–Whitney *U* test was run to determine differences in cytokine concentration between groups. Exact *p*-values corrected for ties were calculated and differences considered significant for *p*-values ≤ 0.05. In case of normally distributed data, an independent samples *t*-test was performed to support the results (data not shown). Patient’s plasma samples were divided into three subgroups, based on parasitaemia. Kendall’s tau b correlation was run to determine the relationship between the analyte concentration and level of parasitaemia. The correlation between the cytokines and parasitaemia was performed by means of a correlation analysis using the nonparametric Spearman correlation (GraphPad Prism, version 9.02 (134)). For multiple testing, the Benjamini–Hochberg adjustment and conservative Bonferroni correction were applied [41].

### 2.5. ANGPTL4 and TNF-α ELISA

Human ANGPTL4 and TNF-α was measured using ELISA after respective dilution of the sample in a reagent dilution buffer following the instructions of the manufacturers (R&D Systems, Minneapolis, MN, USA). Significance was evaluated using the Mann–Whitney *U* test.

### 2.6. RNA Isolation

RNA was isolated using a PureLink RNA Mini Kit (Thermo Fisher Scientific, Bremen, Germany) according to the manufacturer’s instructions. Genomic DNA contamination was removed using the TURBO DNA-free Kit (Invitrogen, Thermo Fisher Scientific, Bremen, Germany) followed by a magnetic bead enzymatic wash using Agencourt RNAClean XP (Beckman Coulter, Krefeld, Germany). The concentration and quality of isolated RNA were assessed using an Agilent 2100 Bioanalyser System with the Agilent RNA 6000 Pico Kit (Agilent Technologies, Ratlingen, Germany). The RNA was sent to BGI (Shenzhen, China), where RNAseq was performed using the Illumina HiSeq 4000 PE100 platform (approximately 11 M PE reads per samples). Reads were quality and adapter trimmed using Trimmomatic [42] and aligned to the human transcriptome by RSEM [43] using Bowtie2 [44] as an aligner. Differential expression was determined using DESeq2 [45].

## 3. Results

### 3.1. Determination of Concentrations of Different Cytokines in Plasmas of Malaria Patients and Healthy Controls

In the first part of this study, the plasmas of 27 patients infected with *P. falciparum* with a parasitaemia between <1% and 11% and of 22 healthy individuals were analysed for the presence of 16 different cytokines. All 27 patients were adult tropical returnees with symptomatic *P. falciparum* malaria (Appendix A). In this study, the pro-inflammatory cytokines IL-1α and IL-1β, the pro- and anti-inflammatory cytokines IL-6, IL-7, IL-11, IL-12 and IFN-β, the anti-inflammatory cytokines IL-1RA and IL-10 and the proinflammatory chemokines CCL3, CCL20, CXCL1, CXCL5, CXCL8 and CXCL10, as well as the growth factor VEGF, were analysed using a customized LEGENDplex assay (Appendix A).

For cytokines IL-6, IL-1RA, IL-10 and IL-11 and chemokines CXCL1, CXCL8, CXCL10, CCL3 and CCL2, significantly higher concentrations were found in plasma samples of malaria patients compared to healthy controls (Figure 1A). For IL-6, IL-1RA, CCL3, CCL20 and CXCL10, the significant difference was already detected at a parasitaemia < 1.0%. For CXCL8, IL-10 and CXCL1, a significantly higher value was found at a parasitaemia ≥ 1% and for IL-11 and CXCL5 only at a parasitaemia > 2.5% (Figure 1A). 

The greatest increase in the amount in the plasmas of malaria patients compared to healthy controls was observed for IL-1RA (H^All^: 1.8 ± 6.9 pg/m, M^All^: 1296 ± 1378 pg/m; 720-fold increase), followed by CXCL1 (H^All^: 3.1 ± 11.7 pg/mL, M^All^: 54.4 ± 103.0 pg/mL; 17.5-fold increase), CCL20 (H^All^: 3.5 ± 5.0 pg/mL, M^All^: 31.8 ± 36.6 pg/mL; 9.1-fold increase) and IL-11 (H^All^: 3.2 ± 8.2 pg/mL, M^All^: 27.1 ± 41.5 pg/mL; 8.5-fold increase) (Figure 1A, Appendix A). For CXCL1, there is a correlation between the amount of cytokine detected and the different levels of parasitaemia (*p* = 0.0013, *r* = 0.5) (Figure 1B). For none of the other cytokines could a correlation with parasitemia be demonstrated. Interestingly, CXCL5 is the only chemokine that was detected at significantly lower levels in plasma of malaria patients than in plasma of healthy controls (Figure 1A). For VEGF, no significant difference was found between patients with malaria infection and the healthy controls, but four malaria plasma samples showed an increase of the VEGF amount, while all remaining individuals had levels beyond the detection limit of the LEGENDplex assay (Figure 1A). The amounts of IL-1 α, IL-1β, IL-7, IL-12 and IFN-β were also below the detection limit of the LEGENDplex assay. 

Thus, for nine of the 16 cytokines examined, a significantly increased amount and for one (CXCL5) a lower amount was found in the plasma of infected individuals compared to healthy controls. Five cytokines were below the detection level of the assay (Figure 1A, Appendix A).

The amount of TNF-α in the plasmas of malaria patients and healthy controls was determined separately by ELISA. On average, significantly less TNF-α is present in the plasmas of malaria patients compared to controls (H^All^: 3770 ± 11,925 pg/mL, M^All^: 75 ± 160 pg/mL; however, this was not significant (*p* = 0.0531) (Figure 2, Appendix A).

### 3.2. Determination of Concentrations of Various Cytokines in the Culture Supernatant of ECs Stimulated with Plasma from Malaria Patients and Healthy Controls

The next step was to investigate whether the plasma of malaria patients and non-infected, healthy individuals have an influence on EC cytokine secretion. It must be mentioned here that, of course, not only cytokines present in plasma, but also various plasmodial antigens and released components, as well as endogenous metabolites, can stimulate ECs. For this purpose, ECs of the brain EC line HBEC-5i were stimulated for six hours with human plasma using a concentration of 25%. Afterwards, the plasma-containing culture supernatant was removed, and the ECs were cultivated in DMEM/F-12 complete growth medium. Cell culture supernatants were collected 48 h after starting the stimulation and the level of cytokines secreted was analysed. Subsequently, the culture medium was removed, and the level of cytokines secreted in the culture supernatant was analysed (Appendix A). Preliminary studies have shown that significant effects could only be measured 48 h after stimulation.

Stimulation of ECs with plasma from malaria patients resulted in significantly increased levels of IL-11, CXCL5, CXCL8, CXCL10 and VEGF in comparison to stimulation of ECs using plasma from healthy controls (Figure 2). In all cases, the measured difference was significant; but, in contrast to the results from the plasma samples, only a 1.5–2.3-fold increase was detected (IL-11: H^All^: 346.4 ± 157.3 pg/mL, M^All^: 526.9 ± 219.3 pg/mL, 1.5-fold; CXCL5: H^All^: 152.5 ± 128.6 pg/mL, M^All^: 263.5 ± 117.8 pg/mL, 1.7-fold; CXCL8: H^All^: 7085 ± 7065 pg/mL, M^All^: 11,681 ± 6360 pg/mL, 1.6-fold; CXCL10: H^All^: 30.4 ± 34.2 pg/mL, M^All^: 69.2 ± 53.6 pg/mL, 2.3-fold; VEGF: H^All^: 112.1 ± 56.2 pg/mL, M^All^: 170.1 ± 59.3 pg/mL, 1.5-fold) (Figure 3, Appendix A).

Again, the amount of IL-1α, IL-1β, IL-7, IL-12 and IFN-β were below the detection limit of the LEGENDplex assay, in this case also of IL-10. In contrast to the CC levels in the plasmas, no differences were detected for the cytokines IL-6, IL-1RA and the chemokines CCL3, CCL20 and CXCL1 in the cell culture supernatants of HBEC-5i cells stimulated with plasma from malaria patients and healthy controls (Figure 3).

If we corrected for multiple testing and included all cytokines with measurable values, most of the reported differences (supernatant and plasma) are still be significant (Figure 1, Figure 3, Appendix A). For the supernatants, there is no different result with either the Benjamini–Hochberg adjustment or the conservative Bonferroni correction. For plasma, CXCL5 and IL-11 fail the Bonferroni correction, while with the Benjamini–Hochberg adjustment CXCL5 proves significant and IL-11 just misses the cut-off (*p* = 0.0457, cut-off = 0.0455).

No TNF-α was detected in the supernatants of endothelial cells after stimulation with plasma from the malaria patients or with plasma from the healthy controls, respectively.

### 3.3. Amount of Secreted Angiopoietin-like Protein 4 (ANGPTL4) in Culture Supernatant of ECs Stimulated with Plasma Derived from Malaria Patients and Healthy Individuals

Studies suggest a synergistic effect of ANGPTL4 and VEGF [46,47]. Therefore, both the plasmas as well as the culture supernatants of the ECs stimulated with plasma were examined for the presence of ANGTPL4 using an ELISA assay. On average, there was less ANGPTL4 in the plasmas of the malaria patients than in the plasmas of the controls (H^All^: 656.8 ± 1108.7 ng/mL, M^All^: 149.9 ± 93.4 ng/mL); however, this was not significant (Figure 4A). When the ECs were stimulated with the plasmas, the reverse was observed. Stimulation with plasma from malaria patients resulted in an increase in the measured amount of ANGPTL4 in comparison to the controls (H^All^: 13.8 ± 3.4 ng/mL, M^All^: 16.4 ± 5.6 ng/mL, *p* = 0.0691). However, the measured amounts were 10–50 times lower than in the plasmas. Considering the different parasitaemia levels separately, only plasma from patients with a parasitaemia of >2.5% has significantly higher levels of ANGPTL4 (20.7 ± 8.5 ng/mL, *p* = 0.0071) in the supernatant compared to the controls (Figure 4B, Appendix A).

### 3.4. Comparative Transcriptome Analyses of ECs Stimulated with Plasma of Malaria Patients and Healthy Individuals

Next, we analysed whether the differences observed on the protein level in the culture supernatant after stimulation of the ECs with plasma of malaria patients could also be found on the RNA level. For this purpose, the HBEC-5i cells were stimulated with the plasma of four malaria patients and of three healthy control individuals for seven hours and subsequently their transcriptomes were analysed. The malaria patients had a parasitaemia between 2.5 and 4% (Appendix A).

After seven hours of stimulation, a significant increase was observed for *il1**β* (*p* = 0.0042), *cxcl1* (*p* = 0.0029), *cxcl5* (*p* = 0.0059) and *angptl4* (*p* = 0.0002) after stimulation with patient plasma. A tendency was only observed for *vegf* (*p* = 0.05). This is due to the measured expression level of the control H8. This deviates significantly from the expression level of the other two controls (expression level: 5820 vs. 860–935). A decrease in expression, albeit non-significant, after stimulation with plasma from malaria patients compared to the controls was observed for *il11* expression (*p* = 0.05), which is in contrast to the LEGENDplex results (Figure 5, Appendix A). Using qPCR analysis for *angptl4* and *vegf*, the difference in gene expression detected after 7 h could no longer be observed 48 h after stimulation (data not shown).

Transcriptome analysis also provides an overall view of the changes in EC gene expression after stimulation with patient plasma. For this analysis, genes with a base mean level ≥40, a differential expression with a fold change ≤0.6/≥1.7 and a padj ≤0.05 were included. Only thirteen genes were identified that were expressed between 1.7- and 3-fold higher after stimulation with the plasma of healthy control individuals compared to stimulation with the patient plasma (Appendix A). On the other hand, 43 genes are expressed between 1.7- and 4.5-fold higher in ECs after stimulation with plasma from malaria patients than after stimulation with plasma from healthy controls (Table 1, Appendix A).

To identify the biological processes in which the proteins encoded by the identified genes but also the cytokines identified by LEGENDplex and ELISA are involved, a gene set enrichment analyses (GSEA) was performed using g:Profiler analysis [48] (Table 1). The g:Profiler analyses shows that within the gene ontology term biological processes (GO:BP), ‘positive regulation of cell migration’, ‘blood vessel development’ and ‘inflammatory response’ are significantly regulated (padj = 7.602 × 10^−5^, 2.542 × 10^−4^ and 2.474 × 10^−4^, respectively). The KEGG pathways ‘rheumatoid arthritis (padj = 4.499 × 10^−4^) and ‘IL-17 signaling pathway’ (padj = 3.436 × 10^−4^) also were found to be upregulated (Table 1). To identify protein–protein interaction networks, a Markov Clustering (MCL) analyses was performed using the program STRING, version 11.0 [49,50]. This analysis yields four clusters, the largest comprising 20 proteins, which are involved in ‘positive regulation of cell population proliferation’ (padj 4.312 × 10^−8^) and ‘tube development’ (padj 1.948 × 10^−5^). The cluster ‘cholesterol metabolic process’ (padj 3.1 × 10^−6^) contains four proteins and the cluster ‘negative regulation of cell differentiation’ (padj 3.4 × 10^−2^) contains three proteins. In addition, an unassigned cluster (three proteins) was predicted (Figure 6, Table 1).

## 4. Discussion

Previously, more than 30 cytokines have been identified whose production is increased due to *P. falciparum* infection and that, as a result, can be detected in higher amounts in plasma of malaria patients compared to healthy controls. For some of them, such as CXCL8 and CXCL10, a correlation with severity of the disease was observed [7,25,30,36,37,51]. Classical immune cells, such as macrophages/monocytes and dendritic cells, are well known for their cytokine production in malaria [22,23,52]. The role of ECs in this context is only fragmentarily understood although they are in constant contact with circulating cytokines and among the first to detect pathogens and they express receptors for pathogen and cytokine recognition (for a review, see [53]). They are the interface between the circulatory system and surrounding tissue, regulating the diapedesis of immune cells (for a review, see [54]) and transporting cytokines from the tissue to the circulatory system [55,56]. Furthermore, they were recently shown to internalize IEs, possibly leading to blood–brain barrier (BBB) breakdown [57]. However, they are also active players in innate and adaptive immune response (for a review, see [54]) and capable of cytokine secretion themselves [58,59]. It is well studied that cytoadhesion of IEs in the capillaries of various organs not only causes blockage of blood flow, which can lead to organ hypoxia and thus organ failure, but also activates ECs. This leads to increased cytokine production, which can induce endothelial dysfunction and thereby contribute to pathogenesis of CM [5,6,7]. An increase in gene expression induced by cytoadhesion has been demonstrated for a number of cytokine-encoding genes [17,18,19]. However, it is not only cytoadhesion of IEs that leads to an increase in cytokine production. This could also be demonstrated for *Plasmodium* antigens. It was shown that hemozoin leads to an increased secretion of CXCL8 and CCL5 from the endothelium [60]. Similarly, isolated *P. falciparum* histones stimulate the production of CXCL8 [61].

There is general agreement that EC activation is important in the pathogenesis of complicated forms of malaria. However, different approaches to identify the underlying mechanisms in EC activation by *P. falciparum* or its metabolites have so far not been able to reach a unified conclusion [62]. One explanation for the divergent results could lie in the tissue-specific variations of the endothelia, which cause different patterns of immune response [63]. However, one should also keep in mind that the immortalised cell lines used, but also the primary endothelial cells, can show different reaction types. In view of the variability in protein expression and chemokine secretion, it is of utmost importance to determine the response of ECs from human brain microvasculature to *P. falciparum* infection if new approaches in the treatment of cerebral malaria are to be pursued.

The aim of the study presented here was to investigate the immunostimulatory potential of plasma samples drawn from *P. falciparum*-infected patients on ECs in absence of adhering IEs. All 27 malaria patients included in the study were adults with symptomatic malaria. However, the severity of infection according to the WHO criteria could only be determined in a subgroup of the samples included in our study (*n* = 17) [64]; for the rest (*n* = 10), the clinical manifestation is unknown. Within these 17 patients, only three patients could be assigned to severe malaria. As a subgroup of three severe malaria patients is too small for statistical analyses with the corresponding corrections for multiple comparisons, it was decided not to distinguish between clinical manifestations in this study.

In plasmas of the 27 travel returnees infected with *P. falciparum* examined in this study, a LEGENDplex assay detected significantly higher concentrations for 9 of the 16 cytokines analysed compared to the corresponding healthy controls, namely, IL-6, IL-1RA, IL-10, IL-11, CCL3, CCL20, CXCL1, CXCL8 and CXCL10. Interestingly, the concentration of CXCL5 in the plasma of malaria patients was below the detected levels in controls. This is consistent with the observation made by a study conducted in Cameroon, which shows decreased serum levels of CXCL5 in *P. falciparum*-infected individuals. The reason for the lower amount of CXCL5 in plasma of malaria patients is not yet clear [35]. A similar observation was made for TNF-a. Again, there was a tendency for greater amounts to be present in the plasmas of the healthy controls compared to the malaria patients, which was nevertheless not significant.

The dysfunction of ECs and the associated development of vascular damage in the brain, resulting in impairment of the BBB, is one of the consequences of a malaria infection. Oggungwan and colleagues demonstrated that sera from malaria patients are able to increase cell permeability in vitro [65]. Increased endothelial permeability associated with malaria has also been shown elsewhere [66,67,68,69]. One trigger of endothelial dysfunction may be stimulation by various cytokines. They could be circulating in the blood stream or produced within the surrounding tissues or by the ECs themselves, acting in an autocrine or paracrine manner. ECs can produce pro-inflammatory and anti-inflammatory cytokines and chemokines as well as growth factors in response to various stimuli, including IL-1α, IL-1β, IL-3, IL-5, IL-6, IL-10, IL-11, CXCL8, CXCL10, IL-11 and VEGF ([70]; for a review, see [71]).

After stimulation of ECs with plasma from malaria patients, we could measure an increased amount in the culture supernatants for IL-11, CXCL5, CXCL8, CXCL10, VEGF and ANGPTL4 (only if plasma of malaria patients with a parasitaemia >2.5% were used) compared to stimulation with plasma from healthy individuals. For all other cytokines examined, the prevalence of a malaria infection in the plasma donor led to no significant differences in cytokine secretion by ECs stimulated with the plasma. This is also the case for TNF-α, which was not detectable in the supernatants of endothelial cells stimulated with both plasma from the control and malaria patients. Furthermore, no expression of the TNF-α coding gene could be detected. This is in contrast to the described increased expression of TNF-α after direct interaction of IEs with endothelial cells [17,18,19].

However, it must be emphasized here that the stimulation experiments were carried out with the immortalised brain endothelial cell line HBEC-5i. Although this exhibits essential features of cerebral ECs, there are also deviations. EC proteins, such as CD51, ICAM-1 and VCAM-1, are presented on the surface, while others, such as CD31, CD36 and CD62E, are absent [72]. In addition, HBEC-5i cells carry chondroitin sulfate A (CSA) as a dominant molecule on its surface [73]. Nevertheless, HBEC-5i exhibits essential features of cerebral EC, including tight junction structures in particular [72].

CXCL8 binds to CXCR1 and CXCR2, the most important receptor for chemotaxis and mostly expressed on neutrophils. In models of ischemic brain injury, blockage of CXCL8 shows neuroprotective effects and leads to a reduction in infarct volume. In traumatic brain injury, elevated CXCL8 levels in cerebrospinal fluids are connected to BBB damage and increased mortality (for a review, see [74]). Additionally, CXCL8 is involved in angiogenesis. It has been shown that recombinant human CXCL8 can induce EC proliferation and is also involved in capillary tube organization [75,76].

As mentioned above, stimulation with plasma from malaria patients resulted in a significantly increased concentration of CXCL8 in the culture supernatant of ECs. Thus, it can be postulated that plasmodial antigens present in plasma might stimulate this secretion, which has also been described in other studies [60,61].

CXCL5, like CXCL8, is important for neutrophil recruitment and activation. The importance of CXCL5 in malaria pathology is unknown. However, CXCL5 has been described to play a role in ischemia–reperfusion-induced injury in human brain microvascular ECs associated with BBB disruption. CXCL5 has been shown to be upregulated in ischemic stroke and this correlates positively with brain injury. In addition, CXCL5 appears to interfere with brain EC function by regulating the p38 MAP kinase signalling pathway [77]. CXCL5-induced impairment of brain endothelial barrier function has also been demonstrated in other contexts [78]. In rats, pretreatment of ECs with IL-10 inhibited CXCL5-mediated cytokine gene transcription [79]. This is consistent with IL-10 functioning as a crucial anti-inflammatory and protective cytokine in experimental cerebral malaria [80,81]. Elevated plasma concentrations of IL-10 are detected in both mild and cerebral malaria, which is compatible with our findings, but for non-survivors of cerebral malaria a decrease in IL-10 levels was shown [27,82]. An inverted ratio in cytokine concentration between the malaria and control group in plasma and supernatant, as observed in CXCL5 and ANGPTL4, does not constitute a contradiction. Instead, it highlights the need to assess cytokine profiles at the cellular level, if aiming to understand the complex interactions taking place in CM. Brain swelling due to the disruption of the BBB and (cytokine-containing) fluid influx was found to occur in 84% of children dying due to cerebral malaria, but only in 27% of the survivors [83]. However, no correlation between peripheral blood cytokine concentrations and the occurrence of brain swelling in these children could be detected, implying a more local event [84]. Cytokine concentrations measured in peripheral blood represent only the systemic effects and are affected by receptor binding, degradation and excretion. The crucial site of cytokine impact is the cell-surrounding micromilieu [85]. HBECs can secrete cytokines in an apical or basolateral direction (for review [86]). Apically released cytokines would be diluted in the circulating blood, creating a locally acting gradient.

VEGF is a key regulator of physiological angiogenesis. VEGF (i) can promote proliferation and migration of ECs; (ii) serve as a survival factor for ECs; and (iii) is known as a vascular permeability factor, based on its ability to induce vascular leakage [87,88,89,90,91]. VEGF is known to bind to vascular endothelial growth factor receptor 1 (VEGFR-1) (Flt-1) and vascular endothelial growth factor receptor 2 (VEGFR-2) (KDR/FlK-1) on ECs, resulting in a mitogen-activated protein kinase (MAPK) signalling cascade [92]. VEGF seems to play a particularly important role in the repair of brain tissue and wound healing ([93], for a review, see [94]). Increased levels of VEGF can be detected in malaria patients and an increased expression of VEGF was also observed in astrocytes of patients who died of CM [90,95]. However, the role that VEGF plays in CM in particular is still not clear. There is evidence of both a protective and a pathogenic influence for VEGF in the pathology in CM (for a review, see [94]). In our study, VEGF could not be detected in any of the 14 samples examined from the healthy individuals and in the malaria patients VEGF could only be detected in four of the 26 plasma samples analysed. This result contrasts with the findings of Furuta and colleagues mentioned above, where elevated VEGF levels were found in malaria patients compared to patients with febrile illnesses or healthy adults [90]. However, Armah and colleagues also found no difference in the VEGF levels between Ghanaian children with CM, severe malaria or not infected with *Plasmodium* [25]. One explanation for these divergent results in malaria research in general might lie within genetic differences. *P. falciparum* is the strongest known force of evolutionary selection in the recent history of humankind. Diverse adaptions led to differences in resistance, reactions and susceptibility to plasmodial infections between ethnic groups and individuals (for a review, see [96]). A different picture emerges for the amount of secreted VEGF in the culture supernatants of plasma-stimulated ECs. Here, we could detect significantly (*p* = 0.0044) higher concentrations in the supernatants of ECs stimulated with plasma from malaria patients compared to controls. In vitro studies also show that parasite antigens (crude extract of IEs) can induce VEGF secretion from, in this case, human mast cell lines [90].

An interplay between VEGF and ANGPTL4 has been described in different diseases, such as obesity and diabetic macular oedema [46,47]. As mentioned above, significantly lower amounts of ANGPTL4 can be detected in the plasma of malaria patients compared to the plasma of healthy individuals. This picture is reversed, however, if one considers the amounts of ANPTL4 in the culture supernatants of ECs stimulated with plasma. Here, just as for VEGF, significantly higher concentrations can be detected in the culture supernatants after simulation with plasma from the malaria patients (with a parasitaemia >2.5%) compared to plasma from the healthy individuals. Both VEGF and ANGPTL4 are proangiogenic molecules. Besides angiogenesis, ANGPTL4 is involved in several other processes, such as lipid metabolism, wound healing, inflammation, and redox regulation (for a review, see [97]). For ANGPTL4, but also VEGF, it has been shown that expression is also strongly increased by hypoxia, thereby leading to induction of angiogenesis [98,99,100].

CXCL10, like VEGF and ANGPTL4, is present in significantly higher concentrations in culture supernatants of ECs stimulated with plasma from malaria patients compared to plasma from healthy individuals. While VEGF and ANGPTL4 have angiogenic and proliferative effects, CXCL10 has angiostatic and anti-proliferative effects [101,102,103]. The important role of CXCL10 is illustrated in a study by Wilson and colleagues. Here, significantly elevated levels of CXCL10 and CXCL4 were found in patients who had died from CM compared to patients who had survived CM or patients with mild malaria [29]. CXCL10 produced by endothelial cells was shown to play a key role in inducing firm adhesion of T cells and preventing cell detachment from the brain vasculature. The induction of CXCL10 was completely dependent on IFN-γ receptor signalling and played a crucial role in mediating the T-cell–endothelial cell adhesion events that initiate the inflammatory processes that damage the endothelium and promote the development of CM [104]. Bodnar and colleagues showed that incubation of ECs with CXCL10 also significantly reduced tube formation [105].

That the angiogenesis of ECs is strongly influenced by the plasma of malaria patients also becomes clear when looking at the differential gene expression after stimulation of ECs with plasma from malaria patients in comparison to healthy individuals (Table 1). In particular, GO terms such as ‘positive regulation of cell migration’, ‘blood vessel/tube development’, ‘negative regulation of cell differentiation’ and ‘inflammatory response’ were significantly upregulated in ECs stimulated with patients’ plasma in comparison to the controls. Based on these results, it can be postulated that there must be a very delicate balance between these molecules to stimulate proliferation of ECs on the one hand and to limit angiogenesis as well as endothelial dysfunction.

## 5. Conclusions

Our results clearly show that not only cytoadhesion of IEs can lead to stimulation of ECs, inducing the production of various cytokines, but also the plasma of malaria patients, specifically, the parasite and host molecules contained therein, which trigger these processes and thus cause a different cytokine profile than the plasma of healthy controls. IL-11, CXCL5, CXCL8, CXCL10, VEGF and ANGPTL4 have been secreted in significantly higher amounts. This is consistent with the pre-existing finding that plasma from malaria patients impairs endothelial barrier integrity in human umbilical vein ECs [65]. We were able to demonstrate the activation of ECs derived from the microvasculature of the human brain and specify their response. However, we did not identify the plasma factors responsible for this effect and thus cannot say whether they are of parasitic or host-specific origin.

## Figures and Tables

**Figure 1 cells-10-01656-f001:**
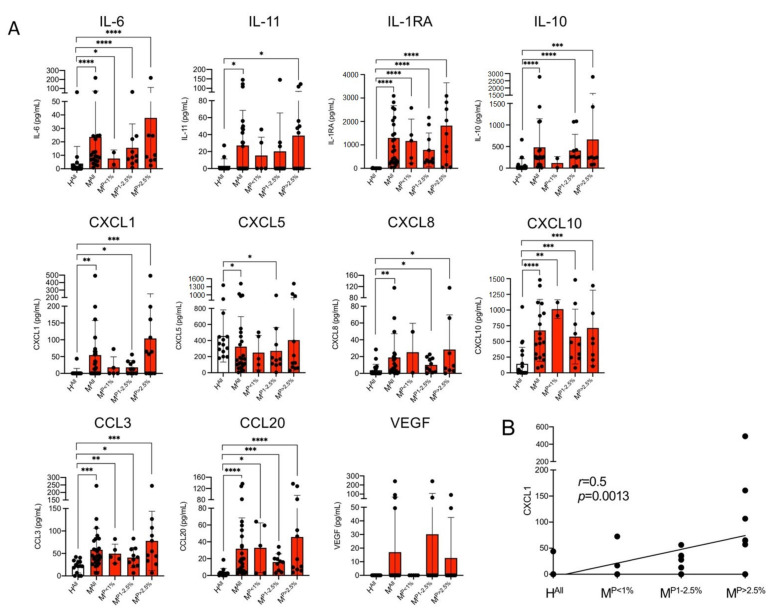
Levels of cytokines in plasma derived from malaria patients (M) and healthy control individuals (H). (**A**) The amount of cytokines was measured with a bead-based LEGENDplex assay (*n* = 13–26, Appendix A). Data are expressed as the mean ± standard deviation (SD). Statistical analyses were performed using the Mann–Whitney *U* test (* *p* < 0.05; ** *p* < 0.01; *** *p* < 0.001; **** *p* < 0.0001). (**B**) For CXCL1, the correlation between the amount of cytokine and parasitaemia was performed by means of a nonparametric Spearman correlation using GraphPad Prism (version 9.0.2 (134)). Abbreviations: Healthy controls (H^All^); malaria patients (M^All^); malaria patients with a parasitaemia < 1% (M^P<1%^), 1–2.5% (M^P1−2.5%^) and >2.5% (M^P>2.5%^).

**Figure 2 cells-10-01656-f002:**
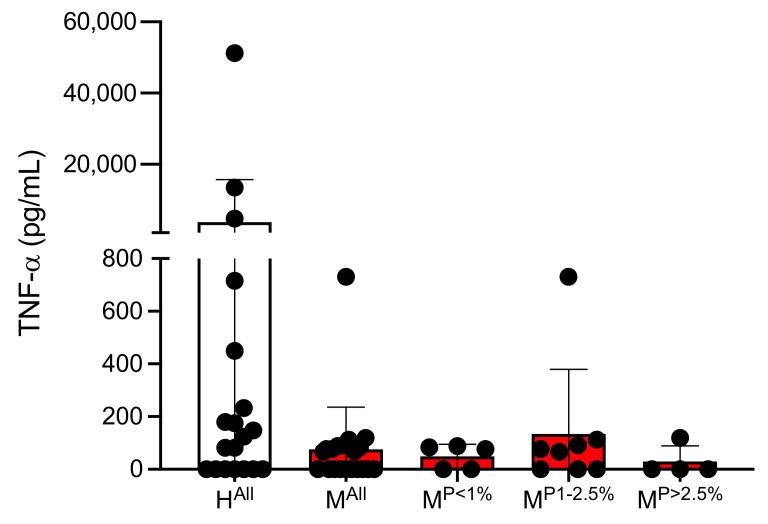
Amount of TNF-α in plasmas of malaria patients and healthy individuals (Appendix A). Abbreviations: Healthy controls (H^All^); malaria patients (M^All^); malaria patients with a parasitaemia < 1% (M^P<1%^), 1–2.5% (M^P1−2.5%^) and >2.5% (M^P>2.5%^).

**Figure 3 cells-10-01656-f003:**
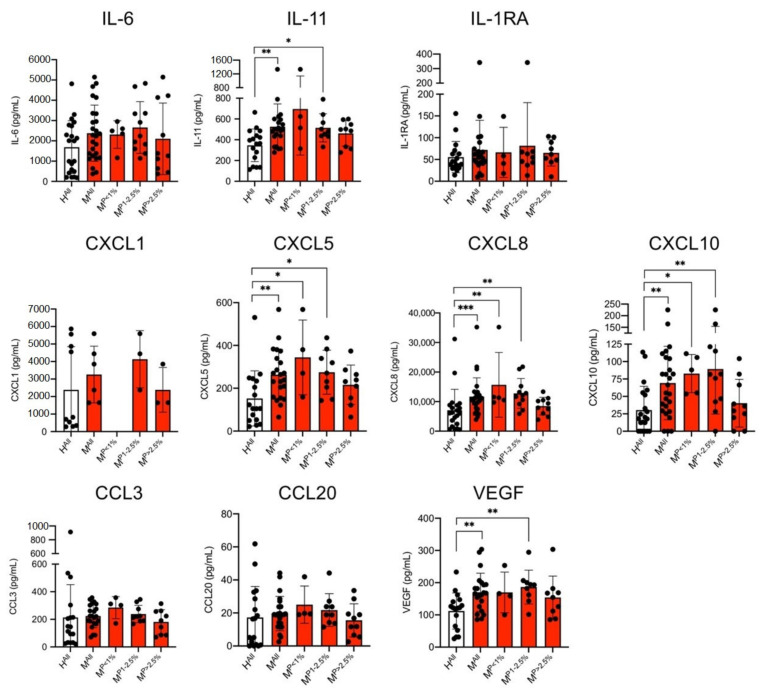
Level of cytokines in the culture supernatants of endothelial cells (HBEC-5i) stimulated with plasma derived from malaria patients (M) and healthy control individuals (H) were analysed with a bead-based LEGENDplex assay (*n* = 6–26; Appendix A). Data are expressed as the mean ± SD. Statistical analyses were performed using the Mann–Whitney *U* test (* *p* < 0.05; ** *p* < 0.01; *** *p* < 0.001). Abbreviations: Healthy controls (H^All^); malaria patients (M^All^); malaria patients with a parasitaemia < 1% (M^P<1%^), 1–2.5% (M^P1−2.5%^) and >2.5% (M^P>2.5%^).

**Figure 4 cells-10-01656-f004:**
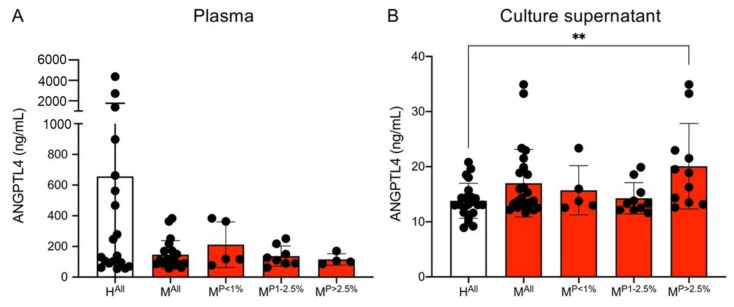
Amount of ANGPTL4 in plasma (**A**) and culture supernatants (**B**) of endothelial cells (HBEC-5i) co-incubated with plasma from malaria patients and healthy individuals. Statistical analyses were performed using the Mann–Whitney *U* test (** *p* < 0.01) (Appendix A). Abbreviations: Healthy controls (H^All^); malaria patients (M^All^); malaria patients with a parasitaemia < 1% (M^P<1%^), 1–2.5% (M^P1−2.5%^) and >2.5% (M^P>2.5%^).

**Figure 5 cells-10-01656-f005:**
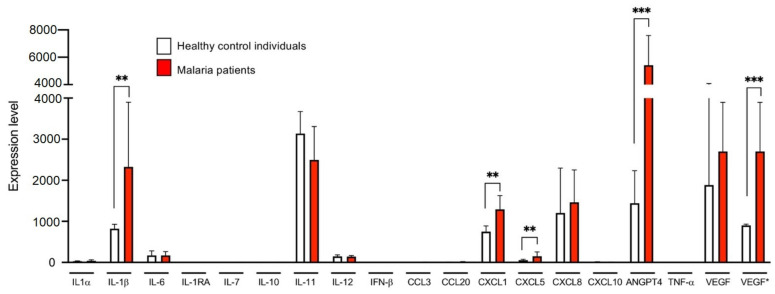
Expression levels of genes coding for the examined cytokines of endothelial cells (HBEC-5i) stimulated with plasma from three healthy control individual and four malaria patients. HBEC-5i cells were incubated for 7 h in the presence of 10% plasma derived from three control individuals and four malaria patients (Appendix A). Subsequently, RNA was isolated from the HBEC-5i cells and a comparative transcriptome analysis was performed (Appendix A). VEGF*: Analysis excluding sample H8. M6, M9, M10, and M11: 4 biological replicates each; H5 and H10: 2 biological replicates each; H8: 1 biological replicate. Statistical analyses were performed using the Mann–Whitney *U* test (** *p* < 0.01; *** *p* < 0.001).

**Figure 6 cells-10-01656-f006:**
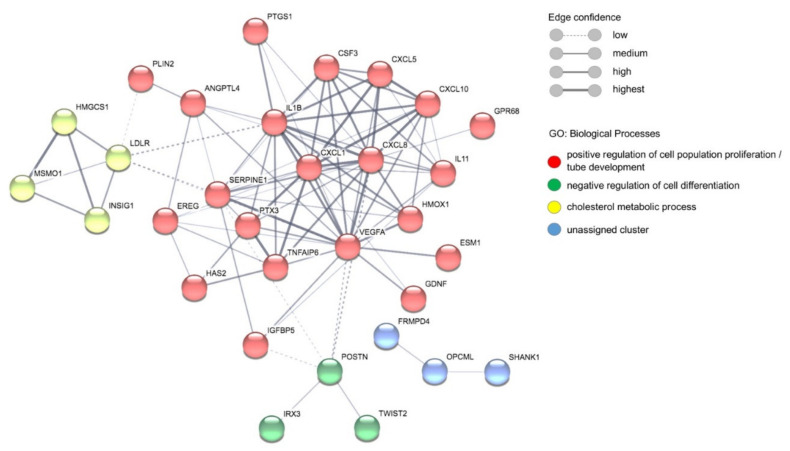
Detection of protein–protein networks through Markov Clustering (MCL) using STRING: functionalprotein association networks [49,50]. Proteins that have not been assigned to a network are not included in the figure. The proteins with a red circle can be assigned to ‘positive regulation of cell population proliferation’ and ‘tube development’, proteins with a green circle can be assigned to ‘negative regulation of cell differentiation’, and proteins with a yellow circle can be assigned to ‘cholesterol metabolic process’ within the gene ontology terms biological processes (GO:BP). Proteins with a blue circle belong to an unassigned cluster.

**Table 1 cells-10-01656-t001:** Genes differentially higher expressed (≥1.7 fold) in endothelial cells incubated with plasma from malaria patients (M6, M9, M10, and M11) compared to incubation with plasma from healthy controls (H5, H10, and H8).

	Gene	Expression Level		Fold Change	padj	GO:BP			KEGG		MCL	Name
		H^5, 10, 8^	M^6, 9, 10, 11^			1	2	3	4	5		
1	CSF3	16	73	4.5	0.016							Colony Stimulating Factor 3
2	ANGPTL4	1442	5419	3.8	4.17 × 10^−6^							Angiopoietin Like 4
3	IL1B	821	2324	2.8	0.019							Interleukin 1b
4	CXCL5	55	149	2.7	0.049							Chemokine cxcl5
5	FOXS1	18	47	2.7	0.002							Forkhead Box S1
6	HMOX1	410	1083	2.6	0.033							Heme Oxygenase 1
7	GDNF	39	98	2.5	0.044							Glial Cell Derived Neurotrophic Factor
8	OPCML	29	71	2.4	3.1 × 10^−5^							Opioid Binding Protein/Cell Adhesion Mol. Like
9	GPR68	23	57	2.4	0.015							G Protein-Coupled Receptor 68
10	INSIG1	1065	2543	2.4	0.002							Insulin Induced Gene 1
11	TNFAIP6	21	49	2.3	0.026							TNF Alpha Induced Protein 6
12	IGFBP5	3873	8901	2.3	3.15 × 10^−7^							Insulin Like Growth Factor Binding Protein 5
13	ESM1	809	1783	2.2	0.022							Endothelial Cell Specific Molecule 1
14	POSTN	920	1922	2.1	5.43 × 10^−5^							Periostin
15	SERPINA9	57	116	2	3.75 × 10^−5^							Serpin Family A Member 9
16	ADTRP	49	100	2	0.023							Androgen Dependent TFPI Regulating Protein
17	MSMO1	1097	2235	2	0.0002							Methylsterol Monooxygenase 1
18	HAS2	1299	2642	2	0.041							Hyaluron Synthase 2
19	LDLR	1491	3030	2	0.0012							Low Density Lipoprotein Receptor
20	SHANK1	25	51	2	0.0036							SH3 And Multiple Ankyrin Repeat Domains 1_2
21	RFX8	38	76	2	0.0031							RFX Family Member 8, Lacking RFX DNA bd.
22	CHST2	305	606	2	0.0012							Carbohydrate Sulfotransferase 2
23	EREG	128	253	2	0.0126							Epiregulin
24	PTX3	6486	12598	1.9	0.00012							Pentraxin 3
25	PTGS1	66	128	1.9	0.0123							Prostaglandin-Endoperoxide Synthase 1
26	RPSAP52	69	133	1.9	1.2 × 10^−7^							Ribosomal Protein SA Pseudogene 52
27	FAM84A	241	455	1.9	0.0012							LRAT Domain Containing 1
28	LAMC2	1258	2329	1.9	0.0341							Laminin Subunit Gamma 2
29	SERPINE1	57674	106566	1.8	0.0002							Serpin Family E Member 1
30	TMEM158	752	1377	1.8	3.63 × 10^−5^							Transmembrane Protein 158
31	FRMPD4	54	97	1.8	0.023							FERM And PDZ Domain Containing
32	B3GNT5	150	267	1.8	0.0099							UDP-GlcNAc:BetaGal Beta-1,3-N-Acetylglucosaminyltransferase 5
33	CAMK1G	111	197	1.8	4.31 × 10^−5^							Calcium/Calmodulin Dependent Protein Kinase IG
34	HMGCS1	1014	1798	1.8	0.016							3-Hydroxy-3-Methylglutaryl-CoA Synthase 1
35	PQLC2L	37	65	1.7	0.0176							Solute Carrier Family 66 Member 1 Like
36	CD93	114	197	1.7	0.00764							CD93 Molecule
37	TWIST2	122	212	1.7	0.0095							Twist Family BHLH Transcription Factor 2
38	IRX3	82	143	1.7	0.0035							Iroquois Homeobox 3
39	CXCL1	749	1293	1.7	0.0017							C-X-C Motif Chemokine Ligand 1
40	RRAD	146	251	1.7	0.0301							Ras Related Glycolysis Inhibitor/Calcium Channel Reg.
41	POU2F2	104	180	1.7	0.0065							POU Class 2 Homeobox 2
42	HSPA1B	3762	6447	1.7	0.0055							Heat Shock Protein Family A (Hsp70) Member 1B
43	PLIN2	2740	4688	1.7	0.0022							Perilipin 2
44 *	VEGF	1884	2700	1.4	ns							Vascular Epidermal Growth Factor
45 *	IL11	3135	2494	0.8	ns							Interleukin 11
46 *	CXCL10	7.3	5.2	0.67	ns							C-X-C Motif Chemokine Ligand 10
47 *	CXCL8	1203	1462	1.2	ns							C-X-C Motif Chemokine Ligand 8

* Differential amount detected in LegendPlex assay. Abbreviations and color code: GO term: Biological Processes (GO:BP): (1) positive regulation of cell migration (padj 8.301 × 10^−6^); (2) blood vessel development (padj 3.601 × 10^−5^); (3) inflammatory response (padj 3.497 × 10^−5^); KEGG pathway: (4) rheumatoid arthritis (padj 1.537 × 10^−5^); (5) IL-17 signalling pathway (padj 2.131 × 10^−5^); MCL-Clustering (GO:BP): red—positive regulation of cell population proliferation (padj 4.312 × 10^−8^) and tube development (padj 1.948 × 10^−5^); green—negative regulation of cell differentiation (padj 3.4 × 10^−2^); yellow—cholesterol metabolic process (padj 3.1 × 10^−6^); blue—unassigned cluster.

## Data Availability

Data is contained within this article and corresponding Appendix A.

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
