# Peer review of "Altered Cytokine Response of Human Brain Endothelial Cells after Stimulation with Malaria Patient Plasma"

_cells, 2021, doi:10.3390/cells10071656_

Round 1
Reviewer 1 Report
Raacke et al adress endothelial response to factors present in plasma of malaria patients. It characterizes the presence of cytokines in plasma of travelers diagnosed with falciparum infection. Later, the authors use these samples to stimulate cytokine production on endothelial cells and overall changes in transcriptomics. The article is timely and well written. Overall properly designed, although some clarifications on the methodology of choice and methodological differences in the manuscript need to be clarified.
Therefore, I recommend the publication of the article with some major revisions, mostly focused into improving a clear description of the methodology and the manuscript technical limitations.
Major:
Methods and results:
- The authors used different methodologies to measure secreted cytokines in plasma and transcription changes on endothelial cells. On the first one, they incubated endothelial cells with 25% plasma and 2.5% heparin for 6h, then removed the plasma and cells are incubated for 42h before cell supernatant collection. On the second method (RNAseq) the authors used a plasma concentration at 10% (without heparin) in a 8h incubation as stated in results (7h as stated in methods?) and directly proceed to RNA extraction? Why did the authors choose such a long incubation time post-plasma treatment on cytokine determination experiments? To what extent these differences in methodology (length of incubation, plasma concentration, addition of heparin) could have accounted for differences shown at transcript and protein level.
Likewise, the authors should clearly mention in results the discrepancies between these two experimental assays (differences in CXCL8, CXCL10, CXCL1, IL11). This is the main novelty of the manuscript and the differences are only briefly and indirectly mentioned in the discussion.
- Figure 1 and 2: Did the authors apply corrections for multiple comparisons on the statistical analysis of cytokines in Figure 1 and 2. As the authors describe, there is only a 1.5-2.3-fold increase and the significance might be lost after correcting for multiple comparisons.
- Figure 4 and supplemental tables S7-9: State whether technical replicates were used on the RNAseq analysis. Legend of figure 4 is unclear and differences and Table S7 seems to suggest it was done. Table S8 and S9 couldn’t be accessed from the supplemental package given.
- Discussion: The authors should be praised for a very extensive discussion and complete bibliography on role of cytokines in cerebral malaria and other neurovascular diseases. However, it fails to mention a few important limitations of the study:
- The authors do not mention the severity of the malaria donors. Did the donors presented any severity criteria? This is important as most of the discussion is focused in associations with cerebral malaria and severe and life-threatening neurovascular diseases.
- Technical limitations: Although the authors state some technical limitations, important limitations are missing. In addition to discrepancies in assay methodology (see above), the authors used an immortalized cell line which is characterized by lack of expression of PECAM/CD31, one of the main markers of endothelial lineage. There are multiple commercial sources of immortalized and primary brain endothelial cells available in the market. Likewise, multiple protocols for brain endothelial cell differentiation from donor-specific iPSC have been developed. The authors discuss that discrepancies in findings in the literature must be due to genetic differences, but failed to discuss that the results obtained in the present study could be cell specific. Although it is unreasonable to validate all the findings in a different cell line/donor in the current manuscript, the authors should mention that the results shown here are limited to a single endothelial cell line, and must be validated in the future with other commercially available alternatives. Likewise, differences in microvascular beds can also be explored in the future in addition to differences in age or donor genetic background.
- Presence of parasite products in plasma. The authors start their study with measurements of plasma cytokines. However, plasma presents other parasite products (PfHRP2, parasite histones, extracellular vesicles, etc) shown to induce endothelial activation. I would suggest to include a sentence acknowledging that in the results (section 3.1 or 3.2). Now, it is only mentioned on the last sentence of the manuscript. This will prevent misleading the reader into the assumption that changes in endothelial cells are solely linked to plasma cytokines.
Minor:
Figure 1: In addition to CXCL1, did other cytokines presented a correlation with parasitemia? Even if not, it is still relevant to mention it on the result section.
Figure 3: Indicate in the figure which graphs correspond to plasma measurements and which ones to endothelial secretion.
Line 310-311. “within the gene ontology (GO) term biological processes (GO:BP)”. Rephrase to “within the gene ontology term biological processes (GO:BP)”.
Table 1 and Figure 5. In the table legend MCL clustering tube development has no color assigned and blue is not described in the legend. Then the colors in Figure 5 do not match colors assigned on Table 1. Please, clarify that and ideally use the same colors for the table and the figure.
Reviewer 2 Report
A descriptive study based on a small cohort of patient sera (27 returning travellers) that looks at cytokines and chemokines in sera of patients infected with Plasmodium falciparum compared to sera from 22 healthy (non-infected) people. This was accompanied by RNA-seq analysis of a brain endothelial cell line "stimulated" with infected versus non-infected sera. Their conclusion was that not only contact of infected red blood cells, but also secreted products in the sera impact on brain endothelial cells.
What was surprising in the study design was that the authors cite TNF 12X in the Introduction and then use a LEGENDplex kit that doesn't measure TNF. Surely, if TNF is worth mentioning 12X in the Introduction it's worth measuring?
In the RNA-seq analysis expression of TNF is not observed, but they do detect expression of a TNF-inducible gene TNFAIP6.
In both cases the absence of TNF is not discussed even though the Discussion has a pronounced tendency to over-interpret the results they did observe.
Author Response
Response to Reviewer 2
A descriptive study based on a small cohort of patient sera (27 returning travellers) that looks at cytokines and chemokines in sera of patients infected with Plasmodium falciparum compared to sera from 22 healthy (non-infected) people. This was accompanied by RNA-seq analysis of a brain endothelial cell line "stimulated" with infected versus non-infected sera. Their conclusion was that not only contact of infected red blood cells, but also secreted products in the sera impact on brain endothelial cells.
What was surprising in the study design was that the authors cite TNF 12X in the Introduction and then use a LEGENDplex kit that doesn't measure TNF. Surely, if TNF is worth mentioning 12X in the Introduction it's worth measuring?
In the RNA-seq analysis expression of TNF is not observed, but they do detect expression of a TNF-inducible gene TNFAIP6.
In both cases the absence of TNF is not discussed even though the Discussion has a pronounced tendency to over-interpret the results they did observe.
As mentioned in the introduction, TNF-a is one of more than 30 cytokines that have been associated with malaria in various studies. All cytokines identified in the various studies were listed, and TNF-a was mentioned as often as some other relevant cytokines were. Unfortunately, it was not possible for us to examine all cytokines with the LegendPlex assay and so we had to limit ourselves to a selection in which TNF-a was unfortunately not included. Since the transcriptome analyses (at least at the time we investigated) also did not give any clear indication of differentially expressed genes that can be assigned to the "TNF signalling pathway", we deliberately did not bring this into the discussion so as not to become too speculative.
Reviewer 3 Report
In the manuscript entitled “Altered cytokine response of human brain endothelial cells after stimulation with malaria patient plasma”, the authors investigated the concentrations of different cytokines in plasmas of malaria patients, as well as the impact of these plasmas on the activation of a brain endothelial cell line. The authors found that significantly higher concentrations of 9 pro-inflammatory cytokines in plasma samples of malaria patients, compared to healthy controls, associated with a lower amount of CXCL5. In addition, the authors found a secretion of ANGPTL4 by EC when stimulated with plasmas from malaria patients. They conclude that these results highlight the importance of the plasma cytokines in the dysregulation of the endothelium in malaria patients. The study design and the methodology are clearly laid out.
The following are some concerns:
- The introduction is too long, as is the discussion which must be reduced. These two sections should be much more concise.
- typo line 252: FN-B
Author Response
Response to Reviewer 3
In the manuscript entitled “Altered cytokine response of human brain endothelial cells after stimulation with malaria patient plasma”, the authors investigated the concentrations of different cytokines in plasmas of malaria patients, as well as the impact of these plasmas on the activation of a brain endothelial cell line. The authors found that significantly higher concentrations of 9 pro-inflammatory cytokines in plasma samples of malaria patients, compared to healthy controls, associated with a lower amount of CXCL5. In addition, the authors found a secretion of ANGPTL4 by EC when stimulated with plasmas from malaria patients. They conclude that these results highlight the importance of the plasma cytokines in the dysregulation of the endothelium in malaria patients. The study design and the methodology are clearly laid out.
The following are some concerns:
- The introduction is too long, as is the discussion which must be reduced. These two sections should be much more concise.
We have shortened the introduction and discussion.
- typo line 252: FN-B
Corrected
Round 2
Reviewer 2 Report
The rebuttal saying:
"Unfortunately, it was not possible for us to examine all cytokines with the LegendPlex assay and so we had to limit ourselves to a selection in which TNF-a was unfortunately not included. "
Is not acceptable, because it suggests that the composition of a commercial kit is determining the research focus. It's entirely feasible to measure TNF directly using other kits.
Furthermore:
Since the transcriptome analyses (at least at the time we investigated) also did not give any clear indication of differentially expressed genes that can be assigned to the "TNF signalling pathway", we deliberately did not bring this into the discussion so as not to become too speculative.
Since the Introduction argues TNF is a player the discordance of the RNAseq data should be discussed and as above, it's entirely feasible to measure TNF expression directly by qRT-PCR.
Author Response
According to the reviewer's wishes, we tested both the plasma samples of the malaria patients and the healthy controls, and the supernatants of the endothelial cells after stimulation with the plasmas for the presence of TNF-a using an ELISA. No TNF-a was detected in any of the supernatants examined.
In some plasmas TNF-a was detected, but there is a tendency (not significant) that more TNF-a is detected in the healthy controls compared to the malaria patients (Figure 2, Supplemtary Table S4).
See lines: 160-164; 223-230, 275-276, 413-415, 432-436
Reviewer 3 Report
Introduction and discussion are still too long, but nevermind.
Author Response
We changed the discussion accordingly and incorporated a small section regarding the TNF-a measurements.
Round 3
Reviewer 2 Report
By making the effort to directly measure TNFa the authors have satisfied my criticism and now the revised manuscript is acceptable for publication.